# Influenza A(H1N1)pdm09 Virus Alters Expression of Endothelial Factors in Pulmonary Vascular Endothelium in Rats

**DOI:** 10.3390/v14112518

**Published:** 2022-11-14

**Authors:** Vladimir Marchenko, Darya Mukhametdinova, Irina Amosova, Dmitry Lioznov, Irina Zhilinskaya

**Affiliations:** 1Smorodintsev Research Institute of Influenza, Russian Ministry of Health, 197376 St. Petersburg, Russia; 2North-Western State Medical University Named after I.I. Mechnikov, 191015 St. Petersburg, Russia; 3Almazov National Medical Research Centre, Russian Ministry of Health, 197341 St. Petersburg, Russia

**Keywords:** Wistar rats, influenza virus A(H1N1)pdm09, endothelial proteins, pulmonary vascular endothelium, plasma, immunohistochemistry, ELISA

## Abstract

Influenza virus infection may cause endothelial activation and dysfunction. However, it is still not known to what extent the influenza virus can dysregulate the expression of various endothelial proteins. The aim of the study is to identify the level of expression of endothelial nitric oxide synthase (eNOS), plasminogen activator inhibitor-1 (PAI-1), and tissue plasminogen activator (tPA) in the pulmonary vascular endothelium, as well as the concentration of PAI-1 and tPA in the blood plasma in Wistar rats. Animals were intranasally infected with rat-adapted influenza A(H1N1)pdm09 virus. The expression of eNOS, PAI-1 and tPA in the pulmonary vascular endothelium was determined by immunohistochemistry; the concentration of PAI-1 and tPA was analyzed by ELISA at 24 and 96 h post infection (hpi). Thus, the expression of eNOS in the pulmonary vascular endothelium decreased by 1.9-fold at 24 hpi and increased by 2-fold at 96 hpi. The expression of PAI-1 in the pulmonary vascular endothelium increased by 5.23-fold and 6.54-fold at 24 and 96 hpi, respectively. The concentration of PAI-1 in the blood plasma of the rats decreased by 3.84-fold at 96 hpi, but not at 24 hpi. The expression of tPA in the pulmonary vascular endothelium was increased 2.2-fold at 96 hpi. The obtained data indicate the development of endothelial dysfunction that is characterized by the dysregulation of endothelial protein expression in non-lethal and clinically non-severe experimental influenza virus infection.

## 1. Introduction

Influenza is an acute respiratory infection that causes significant illness and death in human populations [1]. Human influenza viruses infect mainly the cells of epithelia of conductive airways but can also infect endothelial cells of blood vessels which also express α(2,3)- and α (2,6)-linked sialic acids [2,3,4].

Infection of endothelial cells by influenza A viruses (IAV) may lead to endothelial activation or dysfunction with structural and functional dysregulation [5,6]. One of the most significant consequences of endothelial dysfunction is a reduction in synthesis and/or bioavailability of nitric oxide (NO) by endothelial nitric oxide synthase (eNOS). In addition, endothelial dysfunction can lead to the alteration of the expression of various endothelial proteins and cause vascular hyperpermeability, as well as disruption of intercellular junctions with loss of barrier function [7]. Moreover, endothelial dysfunction is one of the causes of cardiovascular pathology, which confirms the fact that both during and at the end of influenza epidemics, excess mortality is observed in patients in risk groups, in particular, with cardiovascular and pulmonary disorders [8,9].

In this study we investigated the expression of three endothelial proteins: (1) endothelial nitric oxide synthase (eNOS), (2) plasminogen activator inhibitor-1 (PAI-1), and (3) tissue plasminogen activator (tPA).

eNOS plays a significant role in maintaining a healthy cardiovascular system and in its adaptation in cardiovascular diseases [10]. Nitric oxide produced by eNOS is a potent vasodilator, but also has antiviral, anti-atherogenic, antiproliferative and anticoagulatory effects which maintain vascular homeostasis [11,12]. NO also plays a key role in inflammation: under physiological conditions, NO has an anti-inflammatory effect while its overproduction induces inflammation [13].

The important roles of regulation of hemostasis as well as influenza pathogenesis belong to PAI-1 and tPA. PAI-1 (serpin E1) is a serine protease synthesized by various cells including endothelial cells. Under normal physiological conditions, PAI-1 controls the activity of the urokinase plasminogen activator (uPA), tissue plasminogen activator (tPA), and the plasmin and matrix metalloproteinases [14,15]. PAI-1 overexpression is associated with pathological fibrin deposition and thrombotic events, while a decrease in PAI-1 expression may lead to hemorrhage [16]. It is important to note that PAI-1 is an active participant in the influenza virus infection cycle. Infectious activity of the influenza A virus can be diminished by PAI-1, which inhibits cleavage of influenza A virus hemagglutinin [17].

tPA is also a serine protease which converts plasminogen to plasmin needed for dissociation of the fibrin clot. It has been shown that tPA can function as a cytokine, regulating some cell signaling pathways, in particular, activating the NF-κB transcription factor responsible for the development of the inflammatory response [18]. tPA also determines virulence of the influenza virus, since plasmin (released from plasminogen by its cleavage by tPA) is one of the main enzymes that cleaves influenza virus hemagglutinin (HA0 to HA1 and HA2) which increases virus infectivity [19].

Thus, all three endothelial proteins have key roles not only in maintaining vascular homeostasis and hemostasis but they also take part in the IAV reproduction cycle.

The study of endothelial protein expression may reveal the features of endothelial dysfunction in influenza, which will help to improve the pathogenesis-based therapy of influenza virus infection.

## 2. Materials and Methods

### 2.1. Animals

Thirty male Wistar rats weighing 220–250 g were obtained (Preclinical Translational Research Centre, Saint-Petersburg, Russia) and divided into three groups, including two experimental groups and one control group (n = 10). The rats were housed in shoe-box cages with two per cage. Animals were maintained at 22 ± 2 °C and a relative humidity of 50 ± 10% with a 12 h light/dark cycle. All rats received water and food ad libitum. All procedures were approved by the Animal Care and Use Committee of the Smorodintsev Research Institute of Influenza and were carried out in accordance with the principles of humane treatment of animals, regulated by the requirements of the European Convention (Strasbourg, 1986).

### 2.2. Virus

Influenza A/St. Petersburg/48/16 H1N1(pdm09) virus was obtained from the Laboratory of Influenza Evolutionary Variability of the Smorodintsev Research Institute of Influenza (St. Petersburg, Russia). The isolate selection was based on clinical studies of IAV H1N1(pdm09), as a causative agent of hemorrhagic pneumonia [20]. The virus was preliminarily adapted through 9 successive passages through infected lung homogenates of Wistar rats, as described [21]. At the last passage, the infectious activity of the influenza virus was 6.6 lg EID_50_/mL; hemagglutination titer of the virus of 1:1024.

### 2.3. Experimental Influenza Virus Infection

Rats from each experimental group after anesthesia with isoflurane (Abbott, Chicago, IL, USA) were intranasally inoculated with 0.2 mL of rat-adapted influenza A(H1N1)pdm09 virus. Control rats were anesthetized, instilled intranasally with 0.2 mL of alpha-MEM, and necropsied at 24 and 96 h. After necropsy, lungs were aseptically removed and fixed in 10% neutral buffered formalin for 24–48 h at room temperature with subsequent histological processing. The body weight of the rats was recorded before virus or alpha-MEM administration, as well as before necropsy.

### 2.4. Immunohistochemistry

Sections of paraffin-embedded tissue blocks with a thickness of 4–5 μm were placed on poly-L-lysine glass slides (Thermo Fisher Scientific, Waltham, MA, USA). To determine the level of expression of endothelial proteins, monoclonal mouse anti-eNOS antibody (Abcam, Bostin, MA, USA, ab76198), polyclonal rabbit anti-PAI-1 antibody (Abcam, USA, ab66705) and monoclonal mouse anti-tPA antibody (Novus Biologicals, Littleton, CO, USA, AH54-10) were used. Sections were incubated with antibody at a dilution of 1:200 (for PAI-1, tPA) and 1:500 (for eNOS) for 1 h at room temperature in a humid chamber. To detect the studied endothelial protein expression in the autopsy material, the visualization system Envision Flex was used, which included DAB chromogen (Dako, Glostrup, Denmark) applied for 2–3 min. Then, sections were washed three times, counterstained with Mayer’s hematoxylin, dehydrated in isopropyl alcohol, and coverslipped.

Morphometric analysis was carried out in Nis-Elements BR 4.40 (Nikon, Tokyo, Japan) with constant settings using blue channel binarization in automatic mode with constant threshold. For each endothelial protein, the registration threshold was empirically selected: for eNOS—0–105, for PAI-1—0–115, for tPA—0–100. Expression quantification was evaluated by sum intensity parameter—sum intensity in every pixel of the object (region of interest).

### 2.5. Determination of PAI-1 and tPA Concentration in Rat Plasma

Rat plasma was frozen and stored at −20 °C. The concentration of PAI-1 and tPA was analyzed using a commercial kit and standard controls (Abcam, USA). The results of the reaction were taken into account in an enzyme immunoassay analyzer (Anthos, Austria) at a wavelength of 450 nm. The concentration of PAI-1 and tPA in the blood plasma of rats was determined based on the standard curve.

### 2.6. Statistics

Statistical data processing was performed using nonparametric test Mann–Whitney and Kruskal–Wallis using MS Office Excel 2016 and GraphPad Prism 8. Differences were considered statistically significant for *p* values < 0.05.

Descriptive statistics such as standard deviation were used to present the obtained data.

## 3. Results

### 3.1. Body Weights; Clinical Observations

Body weights of the rats were within the normal range, and no statistically significant differences between the groups were observed. No signs of clinical illness were observed in the infected rats over time.

### 3.2. eNOS Expression in Pulmonary Vascular Endothelium

The expression of eNOS in the pulmonary vascular endothelium of rats infected with adapted A(H1N1)pdm09 virus was modulated through the entire study (Figure 1).

The expression of eNOS in the pulmonary vascular endothelium was evaluated by sum intensity parameter (Figure 1D). Thus, eNOS expression decreased by 1.9-fold (7643.7 ± 1802.3) at 24 hpi and increased by 2-fold (30,092.7 ± 10,042.9) at 96 hpi (*p* < 0.05).

### 3.3. tPA Expression in Pulmonary Vascular Endothelium and in Blood Plasma of Rats

The expression of tPA in the pulmonary vascular endothelium of rats infected with adapted A(H1N1)pdm09 virus was increased at 96 hpi (Figure 2C).

Sum intensity of tPA in the pulmonary vascular endothelium was increased by 2.2-fold at 96 hpi compared with the control (*p* < 0.05) (Figure 2D). Since tPA expression over time was at a relatively low level, we additionally analyzed the concentration of tPA in the blood plasma of rats infected with the influenza virus.

The tPA concentration in the rat blood plasma of the infected rats did not change over time (Figure 2E). The tPA concentration was 2393.4 ± 236.41 pg/mL at 24 hpi and 1483.2 ± 418.54 pg/mL at 96 hpi (control 1829.5 ± 699.8).

### 3.4. PAI-1 Expression in Pulmonary Vascular Endothelium and in Blood Plasma of Rats

The expression of PAI-1 in the pulmonary vascular endothelium of rats infected with adapted A(H1N1)pdm09 virus was visually slightly increased at 24 and 96 hpi compared with the control group (Figure 3A–C).

Sum intensity of PAI-1 was increased by 5.23-fold at 24 hpi and increased by 6.54-fold at 96 hpi compared with the control (*p* < 0.05). Since PAI-1 expression over time was at a relatively low level, we additionally analyzed the concentration of PAI-1 in the blood plasma of rats infected with the influenza virus.

The PAI-1 concentration was slightly increased (1104 ± 0.11 ng/mL) at 24 hpi (*p* > 0.05) and was decreased by 3.84-fold (0.24 ± 0.09 ng/mL) compared with the control (0.923 ± 0.33 ng/mL; *p* < 0.05).

## 4. Discussion

In the present study, we have analyzed the expression of endothelial proteins such as eNOS, tPA, and PAI-1 in the pulmonary vascular endothelium as well as the concentration of tPA and PAI-1 in the blood plasma of Wistar rats infected with the adapted influenza virus A/St. Petersburg/48/16 (H1N1)pdm09 over time (at 24 and 96 hpi). All these endothelial proteins not only take part in the regulation of vascular homeostasis and hemostasis but in the IAV reproduction cycle.

As described earlier, the experimental influenza virus infection in Wistar rats is non-lethal with no clinical symptoms, which is consistent with previously obtained data in different inbred lines of rats, particularly Brown-Norwegian and Fischer-344 [22,23]. Nonetheless, the rat-adapted influenza virus A(H1N1)pdm09 was actively reproducing in the pulmonary tissues—at 24 hpi infectivity titer was 6.6 ± 0.2 log EID_50_/mL with significant reduction at 96 hpi—2.2 ± 0.3 lg EID_50_/mL [24].

We also found various histopathological changes in the pulmonary blood vessels of the infected rats including endothelium disintegrated from the media, media disintegrated from the adventitia, thinning of endothelial cells, desquamation, and extravasation of erythrocytes [24].

Analysis of endothelial protein expression was carried out in the endothelium of small blood vessels (microvascular endothelium). As known, pulmonary endothelial cells show significant heterogeneity in structure and function and are divided into two major groups: macrovascular and microvascular endothelial cells [25,26]. However, microvascular endothelial cells play an important role in the pathogenesis of influenza virus infection, as these cells are part of the capillaries connected with the epithelia of the conductive airway [5,27].

Analysis of eNOS expression was modulated over time—decreased by 1.9-fold at 24 hpi and increased by 2-fold at 96 hpi with the control. It is known that a deficiency in eNOS expression in endothelial cells leads to a decrease in NO synthesis and a disruption of regulation of vasodilation and vasoconstriction, which are functional characteristics of endothelial dysfunction [28]. One of the mechanisms for reducing NO synthesis is a decrease in the availability of cofactors in endothelial cells (heme, tetrahydrobiopterin, NADP, FAD and FMN), which leads to the uncoupling of eNOS from the substrate (L-arginine). As a result, eNOS continues to receive electrons from NADP whereby supplying another substrate—molecular oxygen, resulting in the formation of reactive oxygen species (ROS), in particular, superoxide anion [29,30]. ROS can further cause oxidative stress and aggravate endothelial dysfunction [31].

Increased eNOS expression in the pulmonary vascular endothelium at 96 hpi can also be considered as a pathological process. It is known that an increase in eNOS expression leads to an increase in NO production, leading to vasodilation under physiological conditions. However, in the experimental influenza virus infection in Wistar rats at 96 hpi, we registered vasoconstriction of the pulmonary blood vessels instead of vasodilatation [24]. Thus, influenza virus infection, particularly at 96 hpi, may cause eNOS uncoupling.

Changes in tPA and PAI-1 expression in the pulmonary vascular endothelium should be considered in combination. It has been established that tPA, in addition to participating in fibrinolysis, also plays an important role in determining the virulence of the influenza virus, since plasmin (released from plasminogen by tPA cleavage) is also one of the main enzymes that cleaves influenza virus hemagglutinin [19]. In turn, PAI-1 (serpin E1) can reduce the infectious activity of the influenza virus by being able to bind and inhibit the urokinase plasminogen activator (uPA), tissue plasminogen activator and trypsin [32,33]. PAI-1 in an important enzyme in various thrombotic conditions—a decrease in PAI-1 activity is associated with the development of hemorrhagic syndrome, and its increase with the risk of thrombosis [14,34].

The expression of tPA in the pulmonary vascular endothelium was increased at 96 hpi, which can be explained by the fact that, at this time point, infectious activity of the influenza virus is still detected.

The expression of PAI-1 in the pulmonary vascular endothelium was increased both at 24 and 96 hpi which limits the infectious activity of the influenza virus due to the extracellular inhibition of tPA/uPA/trypsin activity. Similar data with PAI-1 overexpression was shown in vitro on A549 cells infected with influenza A/SWN/33 (H1N1) virus at 24 hpi [17]. We can assume that epithelia of the conductive airways with vascular endothelium can inhibit the spread of IAV by an increase in PAI-1 expression. It is also known that the overexpression of PAI-1 with low activity of tPA increases the chance of thrombotic complications, which is also noted in high-risk groups for influenza complications [35,36,37].

An Increase in PAI-1 expression in the pulmonary vascular endothelium and a decrease in PAI-1 concentration in blood plasma is a new aspect of influenza pathogenesis and requires further study. Presumably, a decrease in PAI-1 expression in the blood plasma allows a virus to retain minimal infectious activity, but may also lead to hemorrhage. This is consistent with histopathological changes, in particular, endothelial desquamation and foci of erythrocyte extravasation in the pulmonary tissue of the infected rats [38].

We assume that influenza A viruses (IAV) can cause dysregulation of endothelial protein expression in two ways: (1) by changing the metabolism of infected cells [39]; (2) by viral mimicry of cellular proteins [40].

It is known that IAV, including influenza A(H1N1)pdm09 and A(H3N2) viruses, may affect endothelial cell metabolism by decreasing intracellular NADH-dependent dehydrogenase activity in vitro by 20–40% [41]. It is worth mentioning that not only infectious influenza virus but also two glycoproteins of the influenza virus membrane—hemagglutinin and neuraminidase can affect endothelial cell metabolism in a similar manner.

Previously, we found that many viruses, including the influenza A(H1N1)pdm09 virus, have homology between viral proteins and endothelial proteins (including eNOS, tPA, and PAI-1) [42]. It is quite likely that fragments of viral proteins homologous to endothelial proteins disorganize vascular hemostasis and are released during proteolysis of viral proteins. Proteolytic cleavage of these fragments in viral proteins can be achieved by either cellular or viral proteases. In many viruses, proteases are encoded in their genomes. In addition, structural viral proteins may also have protease activity. For example, such activity has PA subunits of influenza virus RNA polymerase [43].

Another potential mechanism of this dysregulation may be explained by antibody formation to these homologous fragments in viral and cellular proteins which can also provoke autoimmune disease. It is known, for example, that vaccination with the Pandemrix vaccine against the pandemic 2009 H1N1 strain has resulted in a significant increase in the incidence of autoimmune narcolepsy in children, adolescents and adults [44]. Later, it was shown that the hemagglutinin and nucleoprotein epitopes of the influenza A(H1N1)pdm09 virus have homology with hypocretin peptides that can trigger formation of a cross-reactive antibody [45].

Thus, obtained results indicate that the influenza A(H1N1)pdm09 virus causes endothelial dysfunction with alteration in endothelial protein expression (eNOS, tPA, PAI-1) in the pulmonary vascular endothelium in non-lethal influenza virus infection and requires a new approach to influenza therapy, in particular, the administration of vaso-protective medications in combination with etiotropic therapy.

## Figures and Tables

**Figure 1 viruses-14-02518-f001:**
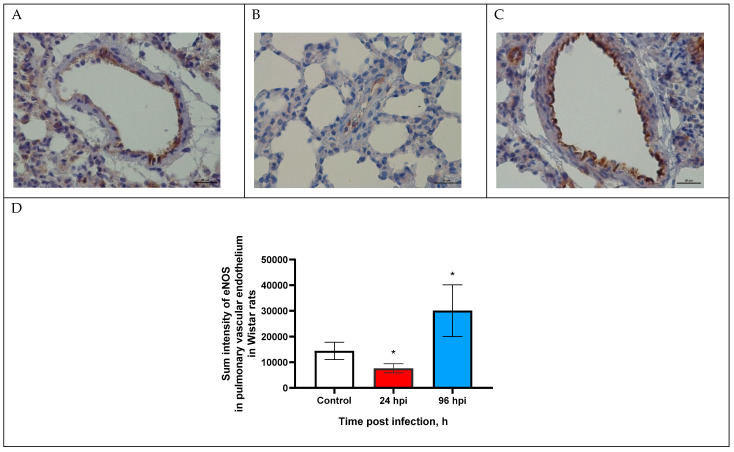
eNOS expression in pulmonary vascular endothelium of control rats (**A**) and infected at 24 (**B**) and 96 h post infection (**C**) with influenza A(H1N1)pdm09 virus using anti-eNOS murine mAbs. Magnification ×400; DAB chromogen staining. (**D**) eNOS sum intensity in pulmonary vascular endothelium of rats infected with influenza A(H1N1)pdm09 virus. Values represent mean ± standard deviation from 30 blood vessels of 10 rats in every group. * *p* < 0.05 versus control group (Kruskal–Wallis test).

**Figure 2 viruses-14-02518-f002:**
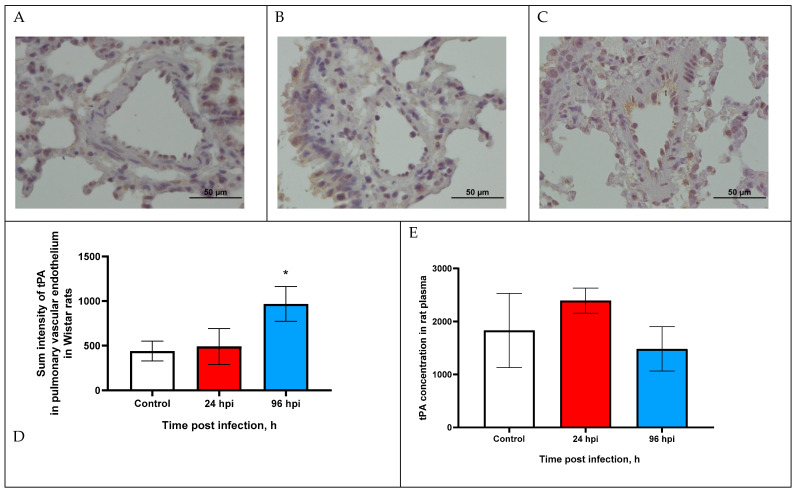
tPA expression in pulmonary vascular endothelium of control rats (**A**) and infected at 24 (**B**) and 96 h post infection (**C**) with influenza A(H1N1)pdm09 virus using anti-tPA murine mAbs. Magnification ×400; DAB chromogen staining. (**D**) tPA sum intensity in pulmonary vascular endothelium of rats infected with influenza A(H1N1)pdm09 virus. Values represent mean ± standard deviation from 30 blood vessels of 10 rats in every group. * *p* < 0.05 versus control group (Kruskal–Wallis test). (**E**) tPA concentration in blood plasma of infected rats infected with influenza A(H1N1)pdm09 virus. Values represent mean ± standard deviation from 10 rats in every group. * *p* < 0.05 versus control group (Mann–Whitney test).

**Figure 3 viruses-14-02518-f003:**
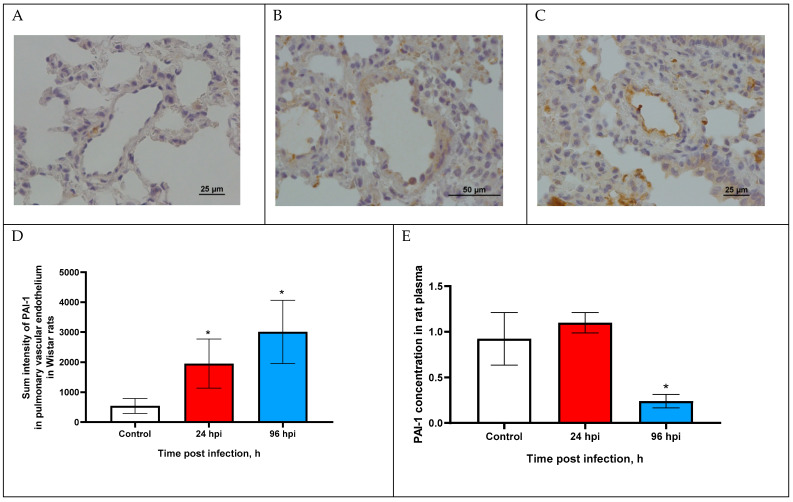
PAI-1 expression in pulmonary vascular endothelium of control rats (**A**) and infected at 24 (**B**) and 96 h post infection (**C**) with influenza A(H1N1)pdm09 virus using anti-PAI-1 rabbit pAbs. Magnification ×400 (**A**,**C**), ×200 (**B**); DAB chromogen staining. (**D**) PAI-1 sum intensity in pulmonary vascular endothelium of rats infected with influenza A(H1N1)pdm09 virus. Values represent mean ± standard deviation from 30 blood vessels of 10 rats in every group. * *p* < 0.05 versus control group (Kruskal–Wallis test). (**E**) PAI-1 concentration in blood plasma of rats infected with influenza A(H1N1)pdm09 virus. Values represent mean ± standard deviation from 10 rats in every group. * *p* < 0.05 versus control group (Mann–Whitney test).

## Data Availability

Not applicable.

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
