# Peer review of "Influenza A(H1N1)pdm09 Virus Alters Expression of Endothelial Factors in Pulmonary Vascular Endothelium in Rats"

_viruses, 2022, doi:10.3390/v14112518_

Round 1

Reviewer 1 Report (Previous Reviewer 2)

Improved a lot based on the first version. However, to be published in this journal, the added discussion part should be revised.

On page 8, "How influenza A virus (IAV) can... with hypocretin peptides that can trigger formation of cross-reactive antibody".

1. Try not to use the active voice "we assume", use the passive voice.

2. It seems like the authors listed "two ways": " 1) changing
metabolism of infected cells [40]; 2) by viral mimicry of cellular proteins [41]". However, the following paragraph is also likely talking about " viral mimicry of cellular proteins". Try to use more explicit examples presenting how IAV changes the metabolism of infected cells.

Author Response

Response to Reviewer 1 

Point 1: Try not to use the active voice "we assume", use the passive voice.

Response 1: Phrase “we assume…” was removed. 

Point 2: It seems like the authors listed "two ways": " 1) changing
metabolism of infected cells [40]; 2) by viral mimicry of cellular proteins [41]". However, the following paragraph is also likely talking about " viral mimicry of cellular proteins". Try to use more explicit examples presenting how IAV changes the metabolism of infected cells.

Response 2: Data on endothelial cell metabolism affected by influenza A virus were added.

Reviewer 2 Report (Previous Reviewer 1)

The authors made significant improvements to the introduction and discussion. More data would strengthen the paper.

Author Response

Response to Reviewer 2 

Point 1: The authors made significant improvements to the introduction and discussion. More data would strengthen the paper.

Response 1: Data on endothelial cell metabolism affected by influenza A virus were added.

All corrections are in green.

This manuscript is a resubmission of an earlier submission. The following is a list of the peer review reports and author responses from that submission.

Round 1

Reviewer 1 Report

The authors use immunochemistry to examine eNOS, PAI-1, and tPA protein expression in rat lung endothelium following influenza infection. The results are descriptive, the technology limiting, and the discussion offers minimal insight.    Recommendations: Methods - indicate the amount of virus used for infection in LD50 units. Results - The authors should compare high dose infection (eg. 1 LD50) with low dose infection  The results do not provide any clinical context. Present some information on clinical signs (eg. weight loss, pulmonary function) eNOS staining appears decreased at 24 hr and increased at 96 hr (Fig 2). But these results need independent validation.  The authors should also validate PAI-1 and tPA immunochemistry results with an another technology. Discussion - Provide some mechanistic insight - does virus directly regulate these proteins in endothelium.  Marchenko and colleagues recently published related work in the same model system. They should consider combining these incremental steps into a comprehensive hypothesis-based story or a review.   References - Include Niethamer, et al., eLife 9:e53072. They describe gene changes and pulmonary endothelial cell heterogeneity following influenza infection.         

Reviewer 2 Report

Vladimir Marchenko, et al, studied the expression of the endothelial factors in H1N1 infected rats. However, to be published in this journal, several points need to be addressed.  

1.       The figures need to be adjusted seriously.

a.       It is too dark to see immunohistochemistry color in Figure 3 and Figure 6.

b.       In Figure 1B, try to switch to another graph that shows bronchiole, as the counterparts in A, and C.

c.       In the legend description of Figures 1, 3, and 5, where is figure panel C?

d.       Since there is only one figure panel in Figures 2, 4, 5, 7, and 8, try to combine with previous immunohistochemistry figures accordingly.

2.       The introduction and discussion need to be improved.

a.       The whole manuscript is talking about endothelial nitric oxide synthase (eNOS), plasminogen activator inhibitor-1 (PAI-1), and tissue plasminogen activator (tPA), so please add more background on these 3 factors in the introduction to strengthen the logical consistency.

b.       Likewise, needs more detail and deep discussion: what is the significance of studying eNOS, PAI-1, and tPA? For example, why “at 96 hpi, eNOS uncoupling is observed”? Any assumptions and related clinical work? Do not use “However, the exact mechanism remains unknown.”

3.       In M & M part, “Rats from each experimental group after anesthesia with isoflurane (Abbott, Chicago, IL, USA) were intranasally inoculated with 0.2 mL of rat-adapted influenza A/St. Peterburg/48/16 (H1N1)pdm09 virus.” Although the authors cited previous publications here, they should list the actual dose (how much LD50) was used for infection, instead of the volume (0.2mL).